# Who Wants to Work More? Multilevel Study on Underemployment of Working Mothers in 22 European Countries

**Milla Salin** [1,*] and **Jouko Nätti** [2]

1   Department of Social Research, University of Turku, 20014 Turku, Finland
2   School of Social Sciences and Humanities, University of Tampere, 33014 Tampere, Finland; jouko.natti@tuni.fi
*   Correspondence: milla.salin@utu.fi

**Abstract:** This study examines underemployment of working mothers in 22 European countries. Underemployed mothers are defined as those who wish to work longer hours than they are currently working. Compared to unemployment and employment in general, the research tradition of underemployment is less established. This article contributes to the existing knowledge on underemployment in two ways. First, it focuses on a specific group of workers: mothers. Secondly, while the vast majority of earlier studies has concentrated on single countries, this study is cross-national. Using data from the 2010/2011 European Social Survey (ESS), a multilevel analysis provides three major findings. First, underemployment exists in all countries examined, but the prevalence varies significantly. Second, the prevalence and depth (i.e., how large is the gap between preferred and current working hours) of underemployment are not necessarily correlated; a high prevalence can be accompanied by shallower underemployment and vice versa. Third, at the individual-level, underemployment particularly hurts mothers who are in a more insecure position in terms of their economic and labor market situation. At the country level, underemployment is related to a poorer economic situation and less-extensive childcare system.

**Keywords:** insider/outsider-theory; labor markets; mothers; multilevel analysis; underemployment; working time

## 1. Introduction

When the performance of labor markets is evaluated, the most common indicators used are the unemployment and employment rates. Low unemployment and/or high employment rate are thought to imply a labor market's ability to function properly (e.g., ILO 2014; EU 2019; OECD 2019) even though they evidently do not indicate that a labor market is functioning efficiently (Glyde 1977; Campbell 2008). Moreover, by focusing solely on unemployment and employment rates, the wider picture in terms of employment quality is missed (Watson 2000).

From the quality perspective, mismatch between workers' desired and actual working time is crucial. When (in)consistency in preferred and actual working times have been studied, the conclusion has been that, at the aggregate level, working longer hours than one prefers is more prevalent than time-related underemployment, which refers to a situation where actual working time is shorter than that preferred[1]. Nevertheless, when the analysis is brought to an individual level, the results show that

---

[1]   In this article, underemployed mothers are those who wish to work longer hours than they are currently working. This definition differs somewhat from definitions of Eurostat (2011) and ILO (2014) as availability to work longer hours and

underemployment is a reality for a significant number of European workers (e.g., Haataja et al. 2011). According to the Eurostat (2017), in 2016, there were nearly 9.5 million underemployed workers in the EU-28 countries.

Hence, underemployment can be seen as representing the under-utilization of the labor force—a dimension that cannot be captured using the unemployment and employment rates (Bonnal et al. 2009). Furthermore, taking underemployment into account enriches our understanding of labor markets' ability to provide sufficient employment to all who want it (ILO 2014).

Underemployment is also an important phenomenon from a policy perspective. A central objective of employment policies in European countries is that all people who want to work should be able to do so (e.g., EU 2019). In order to meet this objective, it is important to understand the mechanisms that create and maintain underemployment. Moreover, such understanding provides insights for the design and evaluation of existing employment and social policies, as well as directions for their future development. Furthermore, underemployment can have various (often negative) economic consequences for individuals who experience it (e.g., Kjeldstad and Nymoen 2011).

The objective of this study is to examine underemployment of working mothers in 22 European countries by answering the following questions: (1) What is the prevalence of underemployed mothers in different countries? and (2) which individual- and country-level factors can explain the differences in underemployment? The data came from the 2010/2011 European Social Survey (ESS). This article contributes to the existing knowledge on underemployment in two ways. First, it focuses on a specific group of workers: mothers. Earlier studies (e.g., Cam 2012) have revealed underemployment's gendered nature: it is more prevalent among women than men. Still, none of the earlier studies have taken working mothers[2] as their point of departure. Mothers are an interesting group because it has been suggested that individual, family, and contextual conditions are especially crucial for them in determining the possibility of working as many hours as one prefers (e.g., Kjeldstad and Nymoen 2012; Salin 2014)[3] Second, the majority of earlier studies have concentrated on single countries, so cross-national studies have been mainly absent (see, however, Haataja et al. 2011). Cross-national examination enables the results of each country to be seen from a broader perspective (i.e., how each country relates to others).

Compared to unemployment and employment in general, the research tradition of underemployment is less established. Hence, no single theory exists to adequately explain underemployment from a social policy perspective. Part of the reason for this is that the majority of earlier studies on underemployment have been conducted by management scholars (McKee-Ryan and Harvey 2011). In this article, underemployment is approached theoretically from the insider/outsider division of primary and secondary labor markets. It has been suggested that labor markets are becoming increasingly divided into primary and secondary labor markets, separating workers between insiders and outsiders. Primary labor markets are considered better insofar as they represent a more stable position, better paid jobs, and more opportunities for advancement, whereas secondary labor markets

---

working time threshold are not included in the definition. A more detailed discussion of the definition of underemployment is presented in Section 2.

[2]   In principle, unemployed mothers who wish to work could also be described as underemployed. However, unemployed are excluded for following reasons. One of the main interests is to analyze how work-related factors, such as employment sector and type of work contract are related to underemployment. Moreover, ESS data is problematic as it includes only 68 unemployed mothers willing to work (plus 16 unemployed not willing to work) who are between 18 and 55 years old and from who there is valid information on all variables needed. Furthermore, these 68 mothers are unevenly distributed across countries. Hence, the results would not have been reliable.

[3]   It would have been interesting to compare underemployment between mothers and fathers. However, fathers' working time patterns in ESS-data prevented this examination. More than 90 percent of working fathers in nearly all countries were working full-time hours. The only exceptions were the United Kingdom and Ireland, where the share was 89 and 83 percent respectively. Moreover, the share and number of cases of underemployed fathers was very low in the majority of countries which would have led to unreliable results. Nevertheless, there were few countries (Finland, France, Ireland, Hungary, Poland and Portugal) where number of cases of underemployed fathers was higher. In these countries, the share of underemployed fathers varied from 18 percent in Finland to 47 percent in Poland.

are characterized as less stable positions, nonstandard work forms, lower wages, and fewer prospects of advancement (e.g., Dickens and Lang 1985; Kalleberg 2000; Lindbeck and Snower 2001; Hudson 2007; Biegert 2014; Chung 2019). Insider/outsider-division is also gendered phenomena: women are more often than men in the outsider position (Schwander and Häusermann 2013; Green and Livanos 2015). In this article, underemployment is understood to represent one dimension of secondary labor markets and is seen as one characteristic of outsider position. The theoretical approach is discussed more detailed in Section 3.

## 2. Defining Underemployment

The problem with the concept of underemployment is that it has no standard definition. Traditionally, underemployment has been divided to time-related and skill-related or visible and invisible underemployment. Time-related/visible underemployment refers to a situation where person is working fewer hours than he or she prefers to. Skill-related/invisible underemployment instead can be measured with skills mismatch, for example person has higher education than is required in his or her job. Another division concerns whether underemployment is defined by objective or subjective criteria (Glyde 1977; McKee-Ryan and Harvey 2011; Bonnal et al. 2009; ILO 2014). In this article, the interest is on subjective time-related underemployment.

The definition of time-related underemployment is usually informed by three criteria. The first concerns the willingness to work more than one currently is. This criterion was the starting point for defining underemployment in the Eurostat (2011) and ILO (2014) studies: for a person to be classified as underemployed, he or she would have had to express his or her perception of insufficient working hours.

The second criterion involves a person's availability to work longer hours (e.g., Haataja et al. 2011). Thus, persons who do not state that they are available to work longer hours are not defined as underemployed. In many cases, this criterion can be justified, but sometimes it is more ambiguous. Its relevance can be questioned, for example, in situations where a mother expresses a preference for longer working hours, but has no alternative childcare options available and thus cannot work more. Many studies (e.g., Lewis 2009) have shown that a mothers' ability to participate in labor markets is constrained by the existing childcare systems. Hence, in order to also include those mothers whose underemployment might be a result of different country-level constraints, in this study, no availability criterion was employed.[4]

The third criterion has to do with the working time threshold. Earlier studies (e.g., Wilkins 2007) have shown that underemployment is more prevalent among part-time workers. The concept of involuntary part-time work is sometimes used to describe a situation where a part-time worker is underemployed. Hence, underemployment and involuntary part-time work are considered to be closely related. They are both seen as forms of underutilization of labor and are more prevalent among women and mothers than men. In addition, the factors explaining underemployment and involuntary part-time work are, to some extent, similar (e.g., Kjeldstad and Nymoen 2012; Hipp et al. 2015; Kauhanen and Nätti 2015).

The idea behind working time threshold stems from the thought that the possible negative economic consequences of underemployment are more severe for workers currently working shorter hours (Wilkins 2007). The difficulty with the working time threshold is that there is no consensus concerning where it should lie. The most often used threshold is 30 h per week, meaning that, for

---

4　The ideal situation would have been to compare mothers' underemployment with and without availability criteria. A comparison between these two measures would have revealed the importance that the availability criteria has for the prevalence of mothers' underemployment. Unfortunately, ESS-data does not include a variable measuring the availability to work longer hours than one currently works and therefore, this kind of comparison was not possible. However, it can be assumed, that the use of availability criteria would decrease the prevalence of underemployment as mothers might see, for example, lack of childcare as a constrain that prevents them from working longer hours (see Kangas and Rostgaard 2007).

a person to be underemployed, he or she cannot currently work more than 30 h per week (e.g., Eurostat 2011). The ILO (2014) suggests that the hour threshold should be decided in consideration of national circumstances. National thresholds can work in single-country studies, but their use poses problems in cross-national research since national circumstances vary across countries. To avoid this problem of comparison, in this article, no hour threshold was used to classify underemployed persons.[5]

Accordingly, underemployed mothers were defined as those who wish to work longer hours than they are currently working.

## 3. Theoretical Background

Among others, (e.g., Lindbeck and Snower 2001) have developed the insider/outsider theory of primary and secondary labor markets. The insider/outsider theory (or dualization theory) is based on the idea that labor markets are divided to two distinct parts: secure primary labor markets with insiders and insecure secondary labor markets with outsiders.[6] The birth of primary and secondary labor markets is traced to the transition from industrial to postindustrial era. While the industrial era was described with stable, secure and full-time employment to nearly all (male) workforce, the postindustrial era is faced with more prevalent unemployment and atypical employment in terms of part-time work and fixed-term work contracts. However, this change has not treated all workers in a similar way, which has led to the division of labor markets. Those who have been hit harder with labor market change and insecurity in terms of employment stability are said to be outsiders working at secondary labor markets. In contrast, those who have been able to stay in more protected and secure employment are said to be insiders working at the primary labor markets (Kalleberg 2000; Schwander and Häusermann 2013; Yoon and Chung 2016).

Therefore, the labor market's position of outsiders is thought to be more insecure and vulnerable than the labor market position of insiders. This less protected position is stated to be evident in several aspects. Previous studies (e.g., Biegert 2014; Chung 2019) have revealed that different work-related, socio-demographic as well as economic factors differentiate insiders and outsiders.

First, different work-related factors are related to the division of insider and outsider workforce. Especially the role of fixed-term contracts, part-time work and working sector has been emphasized. These fixed-term contracts and—particularly involuntary and short—part-time work can be seen as factors related to more insecure outsider position in comparison to full-time work and unlimited contracts. Working in one specific sector is not as such a sign of vulnerable labor market position, but is has been stated that outsider position is more prevalent in service sector than, for example, in the industrial sector (Kjeldstad and Nymoen 2012; Yoon and Chung 2016). Work-related factors have also been argued to play a part in the prevalence of underemployment. Brown and Pintaldi (2006) have shown that underemployment is more common in temporary than continuous positions. The results concerning the sector of employment have varied somewhat depending on the type of measurement. Working in the service sector in general is seen to increase the risk (Nord 1989), but whether this is an issue in the private sector (Cam 2012) or the public one (Caputo and Cianni 2001) remains uncertain.

Second, socio-demographic factors such as education and age are thought to shape the risk of being an outsider. A lower education is argued to be related to outsider position and hence to a more insecure position at the labor market (Schwander and Häusermann 2013; Chung 2018). Education is also related to underemployment. Those with a lower education face a larger risk of being underemployed than the higher educated. A higher education level is seen to provide more opportunities concerning available

---

[5]　Moreover, ESS-data set limitations to the use of working time threshold. In many countries—especially in Eastern European countries—the vast majority of mothers work full-time. Hence, doing analyses only for part-time working mothers would have meant that many countries should have been excluded from the analyses because the number of cases would have been too small. Furthermore, for all other countries the number of cases would also have decreased and the results would not have been reliable. Descriptive statistics of mothers' working times are presented in Appendices A and B.

[6]　In some studies, a distinction is made between employed as insiders and unemployed as outsiders (Lindbeck and Snower 2001).

jobs, which is related to working times. Moreover, the higher educated more often expected to occupy positions where they have more influence on their working time (e.g., Bielenski et al. 2002). The results concerning age and outsider position have been somewhat contradictory. Schwander and Häusermann (2013) stated that young employees have a higher risk of being an outside, whereas some others (e.g., Biegert 2014) have argued that both young and old employees had this risk. Same contradiction is evident in underemployment. Most examinations show that young workers face underemployment more frequently (e.g., Fagan 2001), although others state that underemployment is more concentrated to older workers (Bonnal et al. 2009) or to both young and old workers (Tam 2010).

Thirdly, economic factors are related to insider/outsider-division. Low wages are seen to lead to a more insecure position in economic terms (Schwander and Häusermann 2013; Yoon and Chung 2016). Unfortunately ESS-data has an income-variable which does not take into account the composition of the household, so it is not possible to include income into the analyses. However, a subjective measure of household's economic situation is available. Earlier studies (e.g., Sandor 2011) showed that a household's economic situation affects the risk of being underemployed: workers who perceive their household as suffering from economic problems are more often underemployed. This suggests that in some circumstances, preferred working time patterns are shaped by economic necessity. Nevertheless, it should be noted that economic problems can also be a consequence of an otherwise insecure labor market position.

Finally, characteristics related to family have not been examined concerning insider/outsider- division, but they have been included in studies of underemployment. Single mothers and women whose spouses do not perform paid work experience underemployment more often. In these cases, a higher risk of underemployment could be traced to the necessity of working longer hours in order to make ends meet (e.g., Eurofound 2012). The age of one's children has varying effects on the prevalence of underemployment. According to Fagan (2001), having younger children, and, according to Kjeldstad and Nymoen (2011), having older children, increases the chance that a person is underemployed.

According to insider/outsider-theory, there is variation across countries concerning the division of primary and secondary labor markets. It has been stated that different kinds of country-level characteristics (i.e., institutional configurations) affect the degree of division (Schwander and Häusermann 2013; Hipp et al. 2015; Yoon and Chung 2016). Some earlier studies (e.g., Kjeldstad and Nymoen 2011) on underemployment have also questioned the cross-national variation in the risk of being underemployed and whether different country-level characteristics also influence this risk. One major weakness related to insider/outsider-theory is that the majority of earlier studies have focused on a very limited number of countries: mainly Germany, France, the United Kingdom and occasionally, the Nordic countries have been examined, whereas studies on other countries as well as wide cross-national studies are less common (see however Emmenegger 2009; Schwander and Häusermann 2013; Davidsson and Emmenegger 2013; Chung 2019). Hence, there is hardly any knowledge on insider/outsider-division in Eastern and Southern European countries.

Furthermore, the results of cross-national differences concerning insider/outsider-division are somewhat contradictory. Country-differences have quite often been analyzed by comparing so-called corporatist countries (such as Germany and France) to non-corporatist countries (such as the United Kingdom). Based on these analyses, it has been suggested that the division between insiders and outsiders would be more evident in corporate countries (Palier and Thelen 2010; Biegert 2014). However, Chung (2018) has pointed out that cross-national comparisons concerning the cleavage between insiders and outsiders are not that straightforward: the relatively weaker position of outsiders in some (corporatist) countries might not be a result of their weaker position as such, but due to the fact that insiders' position (in some corporatist) is more better off than in some (non-corporatist) countries. Nevertheless, it can be assumed that there are different country-level factors that are shaping the risk of mother being outsiders or underemployed.

First, factors involving the labor market are expected to be related to outsider position or underemployment. As working at the service sector has been stated to increase the risk of being an

outsider (e.g., Yoon and Chung 2016), it can also be presumed that the importance of service sector as employer (i.e., the share of people working at the service sector) is relevant. This is especially crucial from mothers' perspective as the service sector is a female-dominated sector, a working culture oriented to part-time hours (e.g., Kalleberg 2000). In addition, the strength of trade unions has been one interest in earlier studies. In most cases, trade unions tend to oppose work forms that can be seen to increase precariousness, like part-time work, but there are differences across countries in this respect (e.g., Bolle 1997). Moreover, it has been argued (Palier and Thelen 2010; Davidsson and Emmenegger 2013) that strong trade unions might be more interested in protecting the rights of insiders than outsiders. Hence, it may be that stronger trade unions lead to more evident division between insiders and outsiders.

Secondly, studies on underemployment have questioned whether economic performance and the unemployment situation effect the possibility of working as many hours as one prefers. It has been suggested that in poorer economic conditions and/or in more severe unemployment situations, it might be harder to realize one's preference for long working hours than during better economic and employment periods (e.g., Feldman 1996; McKee-Ryan and Harvey 2011; Bonnal et al. 2009).

Thirdly, when mothers are studied, one characteristic that has to be taken into account concerns childcare arrangements. Earlier studies (e.g., Bielenski et al. 2002) have shown that one reason that mothers work part-time is because their family responsibilities and available childcare options affect mothers' working-time patterns. If alternative childcare (e.g., publicly organized day-care) is not available, mothers may not be able to work as much as they would like (e.g., Salin 2014).

Collectively, earlier literature has revealed interesting points. First of all, underemployment is a reality for many European workers, with women facing it more often than men. What we do not know, however, is what the prevalence of underemployment is among mothers, who constitute a growing (and in many respects, special) group of European workers. Therefore, this study focuses on mothers' underemployment. Furthermore, the existing body of knowledge on underemployment comes mainly from studies that have focused on one or a few countries. Thus, the aim is to widen the perspective and study this phenomenon in 22 European countries. Moreover, based on earlier studies on insider/outsider-division and underemployment presented above it is possible to point to some individual- and country-level factors that increase the risk of underemployment. However, this information is drawn from studies focusing on workers in general. Hence, this study questions whether these same factors can explain mothers' underemployment in a large European sample.

## 4. Research Design

This study examines working mothers' underemployment in 22 European countries by addressing the following questions: (1) What is the prevalence of underemployed mothers in different countries? and (2) which individual- and country-level factors can explain the differences in underemployment? In the first question, the focus is on the differences across countries in the prevalence of underemployed mothers. Moreover, the nature of mothers' underemployment will be examined by studying the depth of underemployment. In the second question, the focus turns to individual- and country-level characteristics that are assumed to be related to cross-national variance in this phenomenon.

Earlier literature gives theoretical and empirical grounds for the present hypotheses. Hypothesis 1 relates to cross-national differences in mothers' underemployment. Studies on insider/outsider-division (e.g., Biegert 2014; Chung 2019) have given contradictory results concerning country-differences and underemployment has not been studied with this wide country selection before. However, cross-national differences are presumed to be found because earlier studies have shown especially short part-time work (one to 19 h per week) to increase the risk of being outsider as well as underemployed (e.g., Haataja et al. 2011; Yoon and Chung 2016). This is an important factor for cross-national differences in mothers' underemployment as the importance of (short) part-time work as a form of mothers' paid work varies across countries, being more prevalent in continental and English-speaking countries (e.g., Hipp et al. 2015). Hence, the following hypothesis is formulated:

**Hypothesis 1 (H1).** *Mothers' underemployment is more prevalent in Continental European and English-speaking countries than in other countries.*

Hypothesis 2 involves the role of different individual-level factors in mothers' underemployment. Earlier studies on insider/outsider-division and underemployment (e.g., Brown and Pintaldi 2006; Biegert 2014) have emphasized that outsiders are in a more insecure labor market position in terms of working part-time and having fixed-term work contracts. Both these factors have special relevance for mothers' underemployment. Working in fixed-term contracts is more prevalent among women than men (e.g., Esping-Andersen 2009). Working short part-time is assumed to have especially crucial importance for mothers' underemployment as working short part-time is more prevalent among mothers than others. However, many mothers also work part-time voluntarily, which might alter the importance of part-time work (Kjeldstad and Nymoen 2012; Hipp et al. 2015). As for the role of employment sector (public vs. private), evidence has been inconclusive (e.g., Caputo and Cianni 2001; Cam 2012), no hypothesis is formulated for that factor.

Moreover, earlier studies on insider/outsider-division and underemployment (e.g., Fagan 2001; Tam 2010; Green and Livanos 2015) have revealed that education and age are related to outsider position and underemployment. Both of these factors can be seen as signs of vulnerability at the labor market: Jobs in the secondary labor markets are more commonly those that do not require high education. Concerning age, young workers are vulnerable in that they usually only have short work history and experience, whereas older workers might face different kinds of age-discrimination.

Furthermore, insecure position in terms of economic factors have shown to be related to outsider position and underemployment (e.g., Sandor 2011; Yoon and Chung 2016). In addition, with family-related factors, such as being a single mother and/or having spouse, working part-time can be seen to reflect insecurity in terms of economic situation. It has been suggested that mothers' preference to work long hours might be driven by the need to make ends meet (e.g., Salin 2014) and hence, would be a sign of economic necessity. As for the role of the age of one's youngest child, evidence has been inconclusive (e.g., Kjeldstad and Nymoen 2011), no hypothesis is formulated for that factor.

Thus, the following hypotheses are formulated:
*Mothers' underemployment is more prevalent among:*

**Hypothesis 2a (H2a).** *those who work in fixed-term contracts.*

**Hypothesis 2b (H2b).** *those working short part-time hours.*

**Hypothesis 2c (H2c).** *low educated.*

**Hypothesis 2d (H2d).** *young and/or old workers.*

**Hypothesis 2e (H2e).** *those who have economic problems.*

**Hypothesis 2f (H2f).** *those without partner.*

**Hypothesis 2g (H2g).** *those whose partner works part-time.*

Hypothesis 3 involves the role of different country-level factors in mothers' underemployment. First, labor market related factors are important also at the country-level (Yoon and Chung 2016). Even though working in the service sector itself cannot be seen as a sign of a more vulnerable labor market position, working in the service sector is related to a higher risk of outsider position and underemployment. For mothers, the role of the service sector can be pronounced, because the service sector is female-dominated (e.g., Esping-Andersen 2009). As for the role of trade unions' strength, evidence has been inconclusive (e.g., Palier and Thelen 2010; Chung 2019), no hypothesis is formulated for that factor.

Secondly, countries' poorer economic and unemployment situation is suggested to make it harder to realize a preference for a longer working time (e.g., Bonnal et al. 2009, see also McKee-Ryan and Harvey 2011). Both of these factors can also be seen to reflect insecurity in terms of economic situation.

Thirdly, highly crucial for mothers' underemployment is the availability of childcare (e.g., Bielenski et al. 2002). If alternative childcare (such as public day-care) is not available, mothers may end up working fewer hours than they prefer to.

Thus, the following hypotheses are formulated.

*Mothers' underemployment is more prevalent in countries with:*

**Hypothesis 3a (H3a).** *higher share of women working in the service sector.*

**Hypothesis 3b (H3b).** *poorer economic conditions.*

**Hypothesis 3c (H3c).** *more severe unemployment.*

**Hypothesis 3d (H3d).** *less available (public) childcare.*

The data used came from the 2010/2011 European Social Survey (ESS). The ESS is a cross-sectional survey covering a range of European countries. Data from 2010/2011 were used because this is the newest batch of data for which the rotating module "Work, family and well-being" includes variables on preferred and normal working times (ESS 2013). The mothers studied were working women aged 18–55 years with at least one child under the age of 18 living in their household. Workers were those who had worked at least one hour during the previous seven days.[7] The final data came from 3446 mothers. The number of cases per country and response rates are presented in the Appendix C.[8]

The dependent variable indicated whether the mother was underemployed or not. No ready-made variable exists in the ESS data; hence, to measure underemployment, information on preferred and normal working times was employed. The variable measuring preferred working time showed the number of hours the person wanted to work weekly ("How many hours, if any, would you choose to work, bearing in mind that your earnings would go up or down according to how many hours you work?"). The normal working time variable expressed how many hours the respondent normally works in a week ("Regardless of your basic or contracted hours, how many hours do/did you normally work a week in your main job, including any paid or unpaid overtime?"). To define whether respondents were underemployed or not, the preferred working time variable was subtracted from the normal working time variable to indicate a gap between preferred and normal working times.

In the analyses, the variable was used as a dummy variable separating those who worked shorter hours than they preferred from those who worked either their preferred hours or longer.[9] A dummy variable separating underemployed from others was chosen as the interest in this study was to compare mothers who are underemployed to other working mothers. Hence the aim was not to make

---

[7]  This kind of cut-point is crude, because it includes mothers with very different working times. Part of the reason for this decision was to keep the number of cases per country as large as possible in order to keep the results reliable. As Appendices A and B show, especially in some Continental European countries, a significant share of mothers work short part-time hours, whereas in Eastern European countries, nearly all mothers work full-time. However, working marginal hours was very uncommon in all countries (not shown in Appendices A and B): only one/a few mothers in Switzerland, Germany, Spain, France, the Netherlands, Norway and the United Kingdom worked between one and four hours. In other countries the minimum working hours varied from five to 20 hours. Nevertheless, this crude cut-point for defining working mothers should be taken into consideration when interpreting the results.

[8]  It should be noted that the response rate of Germany is extremely low. This should be borne in mind when interpreting the results.

[9]  A continuous dependent variable was also tested. However, the distribution of continuous variable was very problematic: as much as 34 percent of mothers worked exactly the same amount of hours as they preferred. Moreover, for example, more than 6 percent worked five and ten hours more than they preferred. Hence, a categorical variable was considered more suitable.

comparisons between mothers who work their preferred hours and longer.[10] The cut-point was were set to zero, indicating that all those who worked at least one hour less than they preferred were defined as underemployed.[11] Descriptive statistics for the dependent variable are shown in Table 1, along with all the independent variables.

**Table 1.** Descriptive statistics for dependent and independent variables (unweighted data).

| Individual-Level Variables | % | n |
|:---|:---:|:---:|
| Whether mother is underemployed or not | | |
| Underemployed | 19.4 | 669 |
| Not underemployed[12] | 80.6 | 2777 |
| Missing | 0 | 0 |
| Age | | |
| 18–33 years | 21.8 | 752 |
| 34–45 years | 60.4 | 2082 |
| 46–55 years | 17.8 | 612 |
| Missing | 0 | 0 |
| Education in years | | |
| 0–10 years | 10.7 | 367 |
| 11–17 years | 71.9 | 2478 |
| 18 and more years | 17.4 | 601 |
| Missing | 0 | 0 |
| Existence of partner and his employment status | | |
| Partner in fulltime work | 77.0 | 2297 |
| Partner in part-time work | 4.2 | 146 |
| Partner is not in paid work | 0 | 0 |
| No partner | 18.8 | 648 |
| Missing | 0 | 0 |
| Age of youngest child | | |
| 0–6 years | 43.3 | 1493 |
| 7–17 years | 56.7 | 1953 |
| Missing | 0 | 0 |
| Household's economic situation | | |
| No problems | 76.9 | 2649 |
| Problems | 23.0 | 792 |
| Missing | 0.1 | 5 |
| Employment sector | | |
| Public | 41.1 | 1418 |
| Private | 58.2 | 2004 |
| Missing | 0.7 | 24 |
| Type of work contract | | |
| Unlimited | 75.9 | 2615 |
| Fixed-term | 14.7 | 505 |
| Missing | 9.5 | 326 |
| Working time | | |
| Short part-time hours | 8.4 | 288 |
| Long part-time hours | 25.9 | 892 |

---

[10] Nevertheless, different kinds of sensitivity analyses were done to test the functionality of the dependent variable and in order to see how the results were affected by including those who worked their preferred hours and those who worked longer than preferred in the same category. When underemployed were compared to only those working longer than they prefer results remained similar. When the underemployed were compared to only those who worked their preferred time, the existence of partner and his employment status as well as employment sector became statistically insignificant. However, it should be noted that in these sensitivity analyses, the number of cases per country was significantly lower, which might affect the reliability of the results. When all three groups were compared, the employment sector became statistically insignificant.

[11] Analyses were also conducted by using alternative cut-points for defining underemployed: three and five hours. The share of underemployed mothers decreased with few percentage points when these cut-points were used, slightly less in Eastern European countries than in other countries. Otherwise, the results remained similar.

**Table 1.** *Cont.*

| Individual-Level Variables | % | n |
|---|---|---|
| Full-time hours | 65.8 | 2266 |
| Missing | 0 | 0 |
| Country-level variables [1] | Mean | St. deviation |
| GDP per capita, dollars and PPPs (World Bank Open Data 2019) | 28,890 | 9323 |
| Female unemployment rate (OECD Database for Short-Term Labour Market Statistics 2019) | 8.48 | 3.74 |
| Trade union density (Eurostat Employment and Unemployment Database 2019) | 30.4 | 20.1 |
| Service sector employment (ILO Industrial Relations Data 2018) | 81.9 | 9.2 |
| Utilization coverage of childcare (OECD Family Database 2019) | 28.5 | 7.18 |
| Duration coverage of childcare (Eurostat Database on Income and Living Conditions 2018) | 30.8 | 7.83 |

[1] Information before centering the variables.

As independent variables, both individual- and country-level factors were used. As demographic factors, age and education were taken into account. Age was measured using three categories: 18–33, 34–45, and 46–55-year-olds. Education indicated years of full-time education in three categories: 0 to 10 years, 11 to 17 years, and those whose education lasted 18 years or more.

Household-related characteristics included the existence of a partner and his employment status, the age of the youngest child, and the economic situation of the household. The role of the partner and his employment status were measured as a variable separating full-time working partner, part-time working partner, partner that is outside the labor market, and respondents who did not have a partner. The category representing the youngest child's age distinguished between those whose youngest child was under seven years old from those whose youngest child was at least seven years old. Evaluation of households' economic situation relied on respondents' subjective perception and was used as a variable separating households that did not have economic problems from those that did.

As work-related factors, sector of employment, type of work contract, and working time were used. Sector of employment indicated whether a person was working in the public or private sector[13], whereas type of work contract indicated whether their work contract was an unlimited or fixed-term one. Regarding type of work contract, the share of missing cases was rather high. This is mainly due to a large portion of missing cases in Greece (20%), Switzerland (18%), Spain (14%) and the Czech Republic (13%). This should be kept in mind when interpreting the results. The duration of working time measured mothers' current working time, separating short part-time (1–19 h a week), long part-time (20–34 h a week) and full-time (35 or more hours a week) working mothers. [14]

Country-level factors concerning the role of economic conditions, unemployment situation, service sector employment, trade union strength, and childcare system were included in the analyses. Economic conditions were measured using mean GDP per capita between the years 2000 and 2011. Data on GDP per capita came from the (World Bank Open Data 2019) and were expressed in thousands of dollars. Unemployment situation was a variable indicating the female unemployment rate as a mean between 2000 and 2011. It was derived from the (OECD Database for Short-Term Labour Market Statistics 2019). Service sector employment was measured using the mean percentage of employed females that were working in the service sector between the years 2000 and 2011. Data came from

---

[12]　In most of the countries, a larger share of mothers in this group were working longer than preferred hours. However, in Bulgaria, Hungary, Norway, Poland and Portugal, working the preferred amount of hours was more prevalent. The share of mothers working longer than preferred varied from 23 percent in Hungary to 63 percent in Croatia. The share of mothers working the preferred amount of hours varied from 22 percent in Germany to 53 percent in Bulgaria.

[13]　This variable is somewhat problematic as the share of mothers working in the public sector is rather high. This is the situation for all countries and not only for mothers, but also for all working women and men.

[14]　It should be noted that information on current working hours was also used to calculate the dependent variable; therefore, they are interrelated.

(ILO Industrial Relations Data 2018)[15] Trade union strength was measured using mean trade union density between the years 2004 and 2011 and the data came from the Eurostat Employment and Unemployment database.[16]

Two factors were used to capture the role of childcare.[17] The first variable measured the "utilization coverage" of childcare systems. It indicated the enrollment rates of children under the age of six in formal care or early education services in 2008. Data came from the (OECD Family Database 2019). The second variable was "duration coverage", which showed the median number of weekly hours of formal care for children under school age in 2011. This data was derived from the Eurostat Database on Income and Living Conditions (2018).

In the analyses, country-level factors were used both as categorized and continuous variables. In the descriptive analyses, country-level factors were used as three-category variables to better illustrate possible bivariate relationships. Countries were divided into three groups based on whether GDP per capita/unemployment/service sector employment/trade union density rate/utilization coverage/duration coverage in a country belonged to the lowest, middle or highest third of countries included in this study. In the explanatory analyses, country-level factors were used as continuous variables.

The research methods were divided into descriptive and explanatory types. Using descriptive statistics, the aim was to analyze the cross-national differences in the prevalence and depth of mothers' underemployment. Cross-tabulation was used to examine the binary relationships between each independent variable and the prevalence of underemployment. A logistic multilevel regression analysis was employed to determine which individual- and country-level characteristics explained the variance in mothers' underemployment. The advantage over a standard logistic regression analysis is that a multilevel model allows the intercept to vary randomly across countries, whereas in standard regression, it is presumed to be the same. A logistic analysis was chosen because the dependent variable had two categories. A step-by-step procedure was employed in the logistic analysis, beginning with an empty model. Next, all individual-level factors were included in the model. After that, country-level characteristics were added to the model[18] (Hox 2002). When the results of the logistic multilevel regression analysis were presented, average marginal effects, statistical significances, standard errors, BIC values, country variance and VPC were reported. For the logistic analysis, continuous independent variables were centered around their grand mean. Moreover, because of a rather high share of missing data (9.5 %) in variable measuring the type of work contract, a category for missing information was included for that variable to keep the number of cases as large as possible. Results for missing information were not interpreted.

One problem that relates to multi-level modeling is the small N problem. The number of macro-level units should be large enough in order to keep the results reliable[19]. Moreover, the number

---

[15] Information on Switzerland was from the year 2011 and it came from (OECD Employment and Labour Market Statistics 2019).

[16] Information on Bulgaria was from 2011; Croatia and Slovakia from 2013; Greece from 2004, 2007 and 2010; Hungary from 2005, 2007 and 2008.

[17] Comparable and up-to-date indicators for childcare institutions were not available, so information on outcomes was used. Employing information on outcomes instead of institutions can cause problems because the demand and supply of childcare do not always correlate with each other.

[18] Random slope and cross-level interaction models were also tested, but none of the random slopes or cross-level interactions were statistically significant. This does not evidently mean that there are not any statistically significant differences to be found. The number of cases altogether (3446) is small to cover 22 countries. In addition, the number of macro-level units (22 countries) is rather small. Hence, it may be that the random slope and cross-level interaction models (with quite many independent variables) were too complicated for the data used (See Hox 2002). The results on the bivariate analyses conducted by country (not shown here) also point to this direction as there were cross-national differences especially in relation to the existence of partner and his employment status and economic problems. Underemployment was particularly prevalent among single mothers in CZ, DE, FR, HU, IE PT and SK and among those having economic problems in many Eastern and Southern European countries.

[19] Because of the small number of macro-level units, some sensitivity analyses (linear models as well as separate analyses for each country and different country groupings) were conducted to see if one country or some country groupings affected the results.

of macro-level variables in one model is restricted by the number macro-level units in the data (e.g., Hox 2002). Hence, country-level variables were divided to three distinct models. Nevertheless, as Bryan and Jenkins (2016) stated, with a relatively small number (fewer than 30) of macro-level units, the reliability of country effects should be handled with caution because the country variance and standard errors for country-level effects tend to be under-estimated and confidence intervals too narrow. These limitations should be borne in mind when interpreting the results. Moreover, when cross-sectional data is used, it is not possible to state whether the independent variable affects the dependent variable or vice versa. Therefore, results should be interpreted as statistical relationships between independent and dependent variables.

## 5. Results

The underemployment rates in European countries (i.e., the share of working mothers that are working shorter hours than they prefer) are presented in Table 2, together with the depth of mothers' underemployment. The results reveal that mothers' underemployment existed in all 22 European countries, but the prevalence varied. Most striking is the situation in Poland, where around 45 percent of working mothers are underemployed. In addition, underemployment in Hungary was more prevalent than in the other countries. This result is interesting in light of the fact that in Poland and Hungary, the labor markets are mainly based on the norm of full-time jobs, and mothers usually work full-time (Lewis 2009). This raises the question of whether, for example, economic reasons may lie behind mothers' preference to work even longer hours than they currently are. This could also be the situation in the Czech Republic, Portugal, and Greece, where about every fourth working mother is underemployed.

**Table 2.** Mothers' underemployment rate (%) and the depth of underemployment (hours per week; median, mean, and standard deviations) in European countries.

| | Underemployment Rate | Depth of Underemployment | | |
| --- | --- | --- | --- | --- |
| | | Median | Mean | St. Deviation |
| PL | 45.9 | −10.0 | −12.5 | 8.38 |
| HU | 35.1 | −10.0 | −13.6 | 7.72 |
| CZ | 27.5 | −5.8 | −7.1 | 4.09 |
| PT | 26.4 | −8.0 | −12.1 | 9.75 |
| GR | 25.5 | −10.0 | −12.1 | 6.42 |
| DE | 24.3 | −10.0 | −10.3 | 6.0 |
| IE | 24.3 | −10.0 | −10.5 | 6.22 |
| FR | 21.1 | −5.0 | −7.6 | 6.28 |
| EE | 20.7 | −20.0 | −23.8 | 16.61 |
| SK | 18.7 | −8.0 | −8.6 | 6.08 |
| BG | 18.5 | −8.0 | −10.0 | 6.56 |
| UK | 17.5 | −6.0 | −9.7 | 8.13 |
| NL | 17.2 | −5.5 | −6.8 | 4.28 |
| NO | 16.6 | −4.9 | −7.1 | 6.66 |
| ES | 14.8 | −7.6 | −10.1 | 6.54 |
| CH | 14.4 | −7.0 | −9.5 | 9.23 |
| SE | 12.2 | −8.0 | −8.8 | 6.42 |
| BE | 11.7 | −8.0 | −8.0 | 6.04 |
| DK | 10.7 | −5.0 | −6.7 | 6.98 |
| FI | 9.3 | −7.0 | −8.1 | 7.37 |
| SI | 7.0 | −10.0 | −10.7 | 6.0 |
| HR | 6.7 | −20.0 | −20.9 | 14.69 |

Underemployment also seems to be rather common in Germany and Ireland, where slightly under 25 percent of working mothers are underemployed. In these countries, economic necessity is probably not the main reason for mothers working less than they prefer. In both of these countries, part-time

work has traditionally been the way mothers work. Moreover, (public) childcare is offered mainly on a part-time basis, which may hinder mothers' ability to work as long as they want (e.g., Lewis 2009).

At the bottom end of the table are countries where mothers' underemployment is less prevalent. In Sweden, Belgium, and Finland, the underemployment rate is around 10 percent, and it is even lower in Slovenia and Croatia. These are all countries where mothers tend to work rather long hours, and in Finland, Slovenia, and Croatia, full-time work is particularly common (Nergaard 2010). In addition, in Sweden, Belgium, and Finland, the extensive childcare system enables mothers to work longer hours (Lewis 2009).

Besides prevalence, the depth of underemployment is important. The possible negative economic consequences are assumed to be more severe in situations where the gap between preferred and current working hours is larger (Bender and Skåtun 2009). In addition, in this respect, the variation across countries is noticeable. The depth of underemployment is clearly most severe in Estonia and Croatia, where the median value is 20 h (i.e., mothers work 20 h less per week than they prefer). Nevertheless, the large standard deviations reveal that in Estonia and Croatia, there is a wide variation among the countries in the depth of underemployment. Croatia is an intriguing case; even though relatively few mothers suffer from underemployment, it hits those who experience it extremely hard. It should be noted, however, that the number of cases in Croatia is rather small. In Germany, Greece, Hungary, Ireland, Poland, and Slovenia, the depth of underemployment accounts for ten hours a week; thus, mothers work approximately two hours less per day than they prefer. Although this is less than in Estonia and Croatia, two hours a day equals a substantial amount of money in terms of salaries, especially in the long run. In the rest of the countries, the depth of underemployment varies between five and ten hours a week.

The results in Table 2 provide only modest support for hypothesis 1. As expected, cross-national differences in underemployment rate and depth of underemployment are evident. However, and more importantly, countries do not cluster together according to corporatist versus non-corporatist division (Biegert 2014) or welfare state regimes (Esping-Andersen 2009). There are Eastern, Southern as well as Continental European countries with high and low underemployment rates. Moreover, highest underemployment rates are found in countries where mothers' part-time work is not that prevalent, which is opposite to hypothesis 1. Nevertheless, rather high underemployment rates are found in some countries (DE, IE) where mothers' part-time work is more common.

The role of different individual- and country-level factors on underemployment is presented in Table 3, which displays the results of the logistic multilevel regression analysis.[20]

Model 1 included all individual-level variables. According to hypotheses H2a and H2b underemployment is more common among mothers with a less secure labor market position. This result accentuates the argument (Yoon and Chung 2016) that underemployment is related to secondary labor market and outsider position. Especially striking is the relationship between mothers' working time and underemployment. Mothers who are working short part-time hours face a 34 percentage points higher probability of underemployment than mothers working long part-time hours and as much as a 62 percentage points higher probability than full-time working mothers. Although this is not a surprising result, it illustrates the specialness of mothers' underemployment, as in many countries, mothers constitute a significant share of those working short part-time hours. Furthermore, the larger probability of underemployment among part-timers clearly amplifies that mothers do not always work part-time voluntarily (See Kjeldstad and Nymoen 2012).

Moreover, this result accentuates the argument that underemployment and involuntary part-time work are closely related to each other (e.g., Kauhanen and Nätti 2015). Nevertheless, mothers' underemployment is not solely a phenomena of part-timers. Altogether, more than 11 percent of full-time working mothers are underemployed (see Appendix D). However, the share varies a lot across

---

[20] Results of bivariate analyses are presented in Appendix D.

countries (not shown in Appendix D): in Eastern European countries, underemployment is related to full-time working mothers, whereas in Continental European countries, underemployment is related to part-time work. The result concerning Eastern European countries is explained by the fact that in these countries, nearly all mothers work full-time (see Appendices A and B). The relatively small number of cases per country prohibited a more specific cross-country analysis on the relationship between underemployment and (involuntary) part-time work. However, it seems that there is cross-country variation in the strength of the relationship between the two as the tradition and role of part-time work in mothers' employment varies across countries (see also Lewis 2009).

Furthermore, as presumed, working in fixed-term work contract increases the probability of underemployment by six percentage points compared to mothers with unlimited work contract. No hypothesis was made concerning the role of sector of employment because earlier studies (e.g., Cam 2012) had delivered contradictory results. In the case of mothers, working in the private sector is related to a higher risk of underemployment.

**Table 3.** Logistic multilevel regression analyses on mothers' underemployment, average marginal effects and statistical significances (standard errors) [1].

| | Model 0 | Model 1 | Model 2 | Model 3 | Model 4 |
|---|---|---|---|---|---|
| Age | | | | | |
| 18–33 years | | −0.01 (0.02) | −0.01 (0.02) | −0.01 (0.02) | −0.01 (0.02) |
| 34–45 years | | −0.01 (0.01) | −0.01 (0.02) | −0.01 (0.02) | 0.01 (0.02) |
| 46–55 years | | ref. | ref. | ref. | ref. |
| Education | | | | | |
| 18 or more years | | −0.01 (0.03) | −0.01 (0.03) | −0.01 (0.03) | −0.01 (0.02) |
| 11–17 years | | −0.01 (0.02) | −0.01 (0.02) | −0.01 (0.02) | −0.01 (0.02) |
| 0–10 years | | ref. | ref. | ref. | ref. |
| Existence of partner and his employment status | | | | | |
| No partner | | 0.03 * (0.02) | 0.04 (0.02) * | 0.04 * (0.02) | 0.04 * (0.02) |
| Part-time working partner | | −0.04 (0.03) | −0.04 (0.03) | −0.04 (0.03) | −0.04 (0.03) |
| Fulltime working partner | | ref. | ref. | ref. | ref. |
| Age of youngest child | | | | | |
| 7–17 years | | 0.03 * (0.01) | 0.03 ** (0.01) | 0.03 * (0.01) | 0.03 ** (0.01) |
| 0–6 years | | ref. | ref. | ref. | ref. |
| Household has economic problems | | | | | |
| Yes | | 0.05 ** (0.02) | 0.05 ** (0.02) | 0.05 ** (0.02) | 0.05 ** (0.02) |
| No | | ref. | ref. | ref. | ref. |
| Employment sector | | | | | |
| Private | | 0.03 * (0.01) | 0.03 * (0.01) | 0.03 * (0.01) | 0.03 * (0.01) |
| Public | | ref. | ref. | ref. | ref. |
| Type of work contract | | | | | |
| Fixed-term | | 0.06 ** (0.02) | 0.06 ** (0.02) | 0.06 ** (0.02) | 0.06 ** (0.02) |
| Unlimited | | ref. | ref. | ref. | ref. |
| Working time | | | | | |
| Full-time hours | | −0.62 *** (0.03) | −0.58 *** (0.04) | −0.59 *** (0.04) | −0.58 *** (0.03) |
| Long part-time hours | | −0.34 *** (0.03) | −0.31 *** (0.03) | −0.31 *** (0.03) | −0.31 *** (0.03) |
| Short part-time hours | | ref. | ref. | ref. | ref. |
| Trade union density rate | | | −0.01 (0.01) | | |
| Service sector employment | | | −0.01 *** (0.01) | | |
| GDP/capita (1000) | | | | −0.01 ** (0.01) | |
| Female unemployment rate | | | | 0.01 (0.01) | |
| Utilization coverage of childcare | | | | | −0.01 ** (0.01) |
| Duration coverage of childcare | | | | | 0.01 (0.01) |
| BIC | 3325.8 | 2936.8 | 2939.7 | 2938.9 | 2934.1 |
| Country-level variance | 0.75 | 0.70 | 0.35 | 0.34 | 0.25 |
| VPC | 0.19 | 0.18 | 0.10 | 0.09 | 0.07 |

[1] Underemployed as a reference category. Statistical significance: * $p < 0.05$; ** $p < 0.01$; *** $p < 0.001$.

Secondly, supporting hypotheses H2e and H2f, underemployment is related to the mothers who are in a more insecure position in terms of economic situation. This further reaffirms that underemployment is related to secondary labor market and outsider position (Schwander and Häusermann 2013). Mothers who state that their household is having economic problems have a five percentage points higher probability of facing underemployment than mothers without economic problems. However, it should be noted that with cross-sectional data, it is not possible to say whether an economically insecure position leads to underemployment or vice versa. The existence of a partner and his labor market status can also be seen as a factor contributing to the economic situation of mothers. The results show that mothers without a partner have a three percentage points higher probability of facing underemployment than mothers with a full-time working partner. Hence, this accentuates the argument that mothers' preferences for longer working times are sometimes driven by economic necessity (see Salin 2014). The situation of single mothers is worrying because the negative consequences of underemployment might be more serious when there is only one earner in the household. Surprisingly, and in contrast to hypothesis H2g, whether one's partner is working full-time or part-time is not related to underemployment. Part of the reason for this might be the fact that so few partners (164 cases) are working part-time.

Finally, in contrast to hypotheses H2c and H2d, model 1 reveals that the prevalence of mothers' underemployment is not primarily shaped by traditional socio-demographic factors. Neither age nor education are related to mothers' underemployment, which suggests that mothers' underemployment is shaped partly by different factors than underemployment in general. Because of contradictory results in earlier studies (e.g., Fagan 2001), no hypothesis was made concerning the age of the youngest child. The result shows that mothers with older children have three percentage points higher probability of being underemployed than mothers of younger children. This may indicate that mothers with younger children prefer shorter working time and/or they do not mind working shorter hours. When children are older, mothers might prefer longer working time, but they are unable to realize their preference. For example, the possibility to change working hours according to one's preferences is not possible in many jobs (See Kjeldstad and Nymoen 2011).

Country-level factors are included in models 2, 3 and 4.

In model 2, factors measuring trade union density and service sector employment were added to the analysis. Contrary to hypothesis H3a, more prevalent service sector employment is related to lower underemployment. When the analysis was done excluding Eastern European countries, the result becomes statistically insignificant. This indicates that there are cross-national differences in the relation between service sector employment and underemployment. The role of trade union density was not hypothesized beforehand because of the contradictory results of earlier studies (e.g., Chung 2019). Model 2 shows that trade union density is not related to mothers' underemployment at all when the roles of other independent variables are taken into account.

Factors related to economic and unemployment conditions were tested in model 3. The results support hypothesis H3b and further reassert the role of economic conditions because a country's poor economic performance is related to a higher prevalence of underemployment. Although the relationship is weak; when GDP per capita increases by one thousand dollars, the probability of underemployment decreases by one percentage point. Thus, in times of poorer economic conditions, it is harder for mothers to realize their preference for long working hours (see Feldman 1996). Contrary to hypothesis H3c, a country's performance concerning the unemployment situation does not affect mothers' underemployment.

Model 4 tested the hypothesis H3d that the unavailability of (public) childcare restricts mothers' ability to work as long as they prefer. The hypothesis is partly supported: availability in terms of utilization is related to the prevalence of mothers' underemployment; in countries with less extensive childcare systems, mothers are more often underemployed than in countries with better childcare. The results show that when the coverage of childcare increases by one percentage point, the probability of underemployment decreases by one percentage point. This supports the argument that when

mothers' underemployment is studied, questions related to the organization of childcare should be taken into account (see also Kjeldstad and Nymoen 2012). However, it should be noted that the childcare coverage variable actually measured the *use* of childcare (Lomazzi et al. 2018). Hence, more precisely, the probability of being underemployed increases by one percentage point when the usage of childcare decreases by one percentage point. Quite surprisingly and contrary to hypothesis, duration coverage of childcare was not related to mothers' underemployment. One reason for this might be the above-mentioned problem of childcare variables: results might have been different if it had been possible to measure the available duration of childcare not from demand, but from the supply-perspective.

The last part of Table 3 presents the random effects. The VPC value indicates that the inclusion of country-level factors in models 2, 3, and 4 reduced the share of unexplained country-level variance. Nevertheless, country-level variance tends to get smaller with relatively smaller number of macro-level units (Bryan and Jenkins 2016). Hence, these results should be treated with caution. The same caution applies to the small standard errors of country-level variables in models 2, 3, and 4.

## 6. Conclusions

This article examined working mothers' underemployment in 22 European countries. The results show that underemployment touches a significant (although varying) share of mothers in all the countries included in this study. The highest levels of underemployment are found in Poland and Hungary, where around 45 and 35 percent of mothers, respectively, want to work longer hours than they currently do. In other countries, the share of underemployed mothers varies between over 25 and under 10 percent. The results reassert the conclusion of earlier studies (e.g., Haataja et al. 2011) that underemployment is a reality for a significant number of European workers. Therefore, it is safe to state that underemployment is an important dimension of the under-utilization of the labor force; this should be taken into account in order to better understand the functioning of labor markets.

Another conclusion is that different dimensions of underemployment can draw a different picture of the phenomenon. In this study, both the prevalence and depth of mothers' underemployment were examined. The results indicate that they do not necessarily go hand-in-hand: a higher prevalence of underemployment does not mean that underemployment is deeper, nor vice versa. Thus, in order to better evaluate the nature of underemployment in different countries, both dimensions—prevalence and depth—should be considered. Knowledge of both dimensions is also vital when considering how social and employment policies should be developed to prevent or reduce underemployment. Possible policy actions can be rather different in countries where prevalence is a larger problem than depth, compared to countries where the situation is reversed.

The theory of insider/outsider division to primary and secondary labor markets (e.g., Lindbeck and Snower 2001; Chung 2019) was employed to explain mothers' underemployment. In many respects, the results confirm that underemployment is related to secondary labor markets and outsider position: mothers who are in a more insecure position in terms of their labor market and economic situation are hit harder by underemployment. From a social policy perspective, this result highlights the problematic nature of the underemployment phenomenon. Nevertheless, to some extent, the factors (not) related to mothers' underemployment are surprising in light of earlier studies (e.g., Schwander and Häusermann 2013; Biegert 2014), indicating that mothers' underemployment is related to slightly different factors than underemployment in general. For example, the role of traditional socio-demographic factors in mothers' underemployment is smaller than hypothesized: younger and/or older age and lower education do not increase a mother's risk of being underemployed. It could be concluded that, at least when mothers' underemployment is considered, economic and work-related factors play a larger role than traditional socio-demographic ones in defining one's risk of being underemployed.

Furthermore, the results indicate that not just individual-level factors, but also country-level factors shape the risk of being underemployed. The importance of economic factors is accentuated

by the indication that mothers in poorer countries are experiencing underemployment more often than those in more affluent countries. The availability of childcare is also related to the risk of underemployment: underemployment is more prevalent in countries where childcare systems are less extensive. This result reasserts the argument that available childcare options affect mothers' working time options (e.g., Sandor 2011; Salin 2014).

This article contributes to the existing knowledge on underemployment in two ways: by concentrating on a specific (and growing) group of European workers (namely mothers) and by employing a wide cross-national research design. Although many interesting aspects of mothers' underemployment were revealed, at the same time, avenues were opened for future research. The rather small number of cases in some countries prevented a more detailed analysis in some respects, for example, in the case of the relationship between underemployment and involuntary part-time work.

Moreover, data-related issues prohibited some interesting analyses that would have extended the knowledge on mothers' underemployment and hence would be important to examine in forthcoming studies. These include analyzing the impact that employing working hour threshold has on mothers' underemployment, examination of possible differences in mothers' and fathers' underemployment as well as analysis of more sophisticated random slope and cross-level interaction models. Another interesting question for prospective studies is related to the diverse role of service sector employment in different countries as the division went between "old" and "new" European (Union) countries.

From a theoretical perspective, it would have been informative to compare mothers' underemployment with and without the availability criteria. Furthermore, in order to better understand the nature of underemployment, more attention should be devoted to a deeper analysis of the "risk-factors" of underemployment; for example, different life-stages and population groups. In addition, it would be interesting to extend the analysis to unemployed mothers who would be willing to work in order to analyze whether the malfunctioning of labor markets has varying consequences for mothers across countries: to underemployment in some and to unemployment in others.

Finally, an intriguing issue concerns the possible (negative) economic consequences that underemployment might have for individuals and families. This question is highly relevant as the results show that the mothers who are in a more insecure position in terms of their economic and labor market situation are hit harder by underemployment. A systematic analysis of underemployment's economic consequences would not only paint a more detailed picture of the performance of labor markets and problems related to them, but also define the state of poverty and the role that paid work plays in it. Moreover, this kind of analysis would allow the evaluation and development of current social and employment policies so that underemployment and its possible economic consequences could be prevented or reduced more efficiently in the future.

**Author Contributions:** Conceptualization, M.S. and J.N.; methodology, M.S. and J.N.; validation, M.S. and J.N.; formal analysis, M.S.; investigation, M.S. and J.N.; writing—original draft preparation, M.S.; writing—review and editing, J.N.

**Funding:** This research received no external funding.

**Conflicts of Interest:** The authors declare no conflict of interest.

## Appendix A

**Table A1.** Mothers' preferred and normal working time (mean hours in a week).

|  | Preferred Working Time | Normal Working Time |
|---|---|---|
| BE | 28.2 | 34.2 |
| BG | 40.0 | 41.7 |
| CH | 21.4 | 27.7 |
| CZ | 36.2 | 43.0 |
| DE | 25.8 | 29.4 |
| DK | 30.5 | 36.2 |

**Table A1.** *Cont.*

|  | Preferred Working Time | Normal Working Time |
|---|---|---|
| EE | 35.6 | 40.5 |
| ES | 31.6 | 37.7 |
| FI | 34.6 | 37.6 |
| FR | 31.7 | 34.7 |
| GR | 36.0 | 40.4 |
| HR | 35.2 | 45.9 |
| HU | 39.0 | 38.7 |
| IE | 25.5 | 27.9 |
| NL | 23.3 | 27.1 |
| NO | 34.2 | 36.3 |
| PL | 37.8 | 39.2 |
| PT | 38.3 | 38.2 |
| SE | 34.3 | 37.2 |
| SI | 36.5 | 43.3 |
| SK | 36.3 | 41.8 |
| UK | 23.9 | 28.0 |

## Appendix B

**Table A2.** Descriptive statistics of mothers' normal working time, hours in a week, % (n).

|  | 1–19 h | 20–34 h | 35 and More Hours |
|---|---|---|---|
| BE | 7.6 (13) | 35.7 (61) | 56.7 (97) |
| BG | 0 (0) | 6.2 (8) | 93.8 (121) |
| CH | 25.2 (28) | 45.9 (52) | 27.9 (31) |
| CZ | 0.8 (1) | 5.3 (7) | 93.9 (123) |
| DE | 19.1 (40) | 45.0 (94) | 35.9 (75) |
| DK | 3.4 (6) | 28.7 (51) | 68.0 (121) |
| EE | 3.1 (5) | 9.2 (15) | 87.8 (144) |
| ES | 5.1 (7) | 19.6 (27) | 75.4 (104) |
| FI | 2.7 (4) | 17.9 (27) | 79.5 (120) |
| FR | 4.6 (9) | 26.9 (53) | 68.5 (135) |
| GR | 1.9 (3) | 21.7 (34) | 76.4 (120) |
| HR | 0 (0) | 3.1 (3) | 96.9 (95) |
| HU | 2.6 (3) | 13.7 (16) | 83.8 (98) |
| IE | 18.8 (35) | 46.2 (86) | 35.0 (65) |
| NL | 21.8 (45) | 55.8 (115) | 22.3 (46) |
| NO | 6.1 (12) | 20.7 (41) | 73.2 (145) |
| PL | 5.5 (7) | 14.8 (19) | 79.7 (102) |
| PT | 6.2 (7) | 9.7 (11) | 84.1 (95) |
| SE | 1.2 (2) | 34.3 (59) | 64.5 (111) |
| SI | 0 (0) | 6.3 (8) | 93.8 (120) |
| SK | 2.3 (3) | 4.6 (6) | 93.1 (121) |
| UK | 24.8 (58) | 42.3 (99) | 32.9 (234) |

## Appendix C

**Table A3.** Number of working mothers and response rate per country based on ESS data.

|  | n | % | Response Rate, % |
|---|---|---|---|
| BE | 171 | 5.0 | 53 |
| BG | 129 | 3.7 | 81 |
| CH | 111 | 3.2 | 54 |
| CZ | 131 | 3.8 | 70 |
| DE | 209 | 6.1 | 31 |
| DK | 178 | 5.2 | 55 |

**Table A3.** *Cont.*

|  | **n** | **%** | **Response Rate, %** |
|---|---|---|---|
| EE | 164 | 4.8 | 56 |
| ES | 138 | 4.0 | 69 |
| FI | 151 | 4.4 | 59 |
| FR | 197 | 5.7 | 47 |
| GR | 157 | 4.6 | 66 |
| HR | 98 | 2.8 | 54 |
| HU | 117 | 3.4 | 49 |
| IE | 186 | 5.4 | 65 |
| NL | 206 | 6.0 | 60 |
| NO | 198 | 5.7 | 58 |
| PL | 128 | 3.7 | 70 |
| PT | 113 | 3.3 | 67 |
| SE | 172 | 5.0 | 51 |
| SI | 128 | 3.7 | 64 |
| SK | 130 | 3.8 | 75 |
| UK | 234 | 6.8 | 56 |
| *ALL* | *3446* | *100* | - |

## Appendix D

**Table A4.** Individual- and country-level factors related to mothers' underemployment (%).

|  | **Underemployed** | **Not Underemployed** | **n** [1] | $X^2$ (*p*-Value) |
|---|---|---|---|---|
| **Individual-Level Variables** | | | | |
| Age | | | | 0.07 (0.965) |
| 18–33 years | 18.9 | 81.1 | 715 | |
| 34–45 years | 19.3 | 80.7 | 2116 | |
| 46–55 years | 19.1 | 80.9 | 677 | |
| Education in years | | | | 8.8 (0.012) |
| 0–10 years | 23.6 | 76.4 | 386 | |
| 11–17 years | 19.3 | 80.7 | 2529 | |
| 18 or more years | 16.0 | 84.0 | 612 | |
| Existence of partner and his employment status | | | | 10.7 (0.014) |
| Fulltime working partner | 17.9 | 82.1 | 2439 | |
| Part-time working partner | 18.2 | 81.8 | 159 | |
| No partner | 22.8 | 77.2 | 557 | |
| Age of youngest child | | | | 4.8 (0.029) |
| 0–6 years | 17.5 | 82.5 | 1472 | |
| 7–17 years | 20.4 | 79.6 | 2054 | |
| Household's economic situation | | | | 44.6 (<0.001) |
| No problems | 17.0 | 83.0 | 2737 | |
| Problems | 28.0 | 72.0 | 783 | |
| Employment sector | | | | 6.5 (0.011) |
| Public | 17.0 | 83.0 | 1456 | |
| Private | 21.0 | 79.0 | 2047 | |
| Type of work contract | | | | 53.8 (<0.001) |
| Unlimited | 17.0 | 83.0 | 2676 | |
| Fixed-term | 31.0 | 69.0 | 515 | |
| Working time | | | | 345.8 (<0.001) |
| Short part-time hours | 52.8 | 47.2 | 305 | |
| Long part-time hours | 27.2 | 72.8 | 903 | |
| Full-time hours | 11.6 | 88.4 | 2317 | |

**Table A4.** *Cont.*

|  | Underemployed | Not Underemployed | n [1] | X$^2$ (*p*-Value) |
|---|---|---|---|---|
| **Country-Level Variables** | | | | |
| GDP per capita (dollars and PPPs) | | | | 23.7 (<0.001) |
| Lowest third | 23.3 | 76.7 | 1339 | |
| Middle third | 17.1 | 82.9 | 1114 | |
| Highest third | 16.2 | 83.8 | 1073 | |
| Unemployment rate | | | | 20.1 (<0.001) |
| Lowest third | 15.6 | 84.4 | 1442 | |
| Middle third | 21.6 | 78.4 | 921 | |
| Highest third | 21.7 | 78.3 | 1163 | |
| Trade union density | | | | 35.1 (<0.001) |
| Lowest third | 22.6 | 77.4 | 1271 | |
| Middle third | 21.2 | 78.8 | 1080 | |
| Highest third | 13.7 | 86.3 | 1175 | |
| Service sector employment | | | | 26.6 (<0.001) |
| Lowest third | 23.3 | 76.7 | 1339 | |
| Middle third | 17.8 | 82.2 | 1185 | |
| Highest third | 15.4 | 84.6 | 1003 | |
| Utilization coverage of childcare | | | | 22.8 (<0.001) |
| Lowest third | 22.7 | 77.3 | 1383 | |
| Middle third | 18.5 | 81.5 | 1201 | |
| Highest third | 14.9 | 85.1 | 942 | |
| Duration coverage of childcare | | | | 2.1 (0.089) |
| Lowest third | 20.3 | 79.7 | 1219 | |
| Middle third | 18.8 | 81.2 | | |
| Highest third | 19.2 | 80.8 | 915 | |

[1] Numbers based on weighted data. Therefore, the number of cases is not comparable to Table 1. Moreover, the number of cases varies among variables because missing cases were omitted from the analysis.

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
