# Peer review of "Who Wants to Work More? Multilevel Study on Underemployment of Working Mothers in 22 European Countries"

_socsci, doi:10.3390/socsci8100283_

Round 1

Reviewer 1 Report

The article's objective is clearly specified and it is of scientific interest, given that the topic of underemployment among working mothers specifically - and not just women - has received little attention. Its cross-national nature and its multilevel approach also make it a novel and relevant contribution to previous research. Issues related to the definition of underemployment are well discussed and choices are properly justified. The research design is sound and the different hypotheses are clearly delimited from each other and well-rooted in earlier research. The methods, data and operational indicators used are adequate given the purpose of the article, and also very clearly presented. The authors provide descriptive statistics and thorough information on the variables used. The type of multilevel regression used is well-suited to the purpose of the study, the authors also conduct sensitivity analysis when pertinent, and bear in mind potential methodological limitations when interpreting results. The multilevel approach allows, furthermore, for many interesting results and a nuanced, multidimensional picture in terms of findings, which are interpreted with caution when necessary.

The article is also extremely well written, in clear and correct English, and it is easy for the reader to follow. All in all, it constitutes a solid, well-designed and analyzed contribution to the existing literature on underemployment and working mothers' employment.

Just some minor suggestions:

-For clarity purposes, it would be desirable to specify the definition of underemployment used in the paper as soon as possible, also in the abstract. Issues related to the lack of an standard definition could perhaps also be briefly mentioned in the intro (although they are appropriately dealt with in section 2).

-The specific mechanism linking underemployment to the insider/outsider division, and the one linking such relation to the topic of the paper should also be specified clearly as early as possible (p.2), even though it is well discussed in section 3.

-At the end of section 3, it is stated: "Moreover, based on earlier studies it is possible to point to some individual- and country-level characteristics that increase the risk of underemployment". It might be good to point out that reference is being made to those already noted in the same section.

Reviewer 2 Report

This multilevel study examines underemployment among mothers in 22 European countries.  The results show that underemployment in this group varies by country and that within countries, mothers in less advantaged labor market situations are more likely to be underemployed than others.

               Underemployment is an interesting and important issue and x-national comparisons are especially valuable.  However, while the topic and data are strengths, I have some major concerns about numerous aspects of the research.

I would have liked to see a theoretical and substantive foundation for the project.  The author says that underemployment has been studied less than unemployment, which is true, but it is still important to provide a theoretical framework to motivate the study.  Also, it was not particularly clear why only mothers were studied.  If the question posed were about whether motherhood mattered for unemployment, it would be useful to understand how mothers compared to fathers.  The results suggest that mothers who are disadvantaged in terms of skills and labor market position experience higher levels of underemployment, but that is also likely to be the case among fathers as well.  If so, it is unclear whether we have learned anything about motherhood per se and its relationship to underemployment.  Overall, the paper needs a more theoretically driven and compelling rationale for the study. One result of this is that the hypotheses are not as well-grounded as one would expect. Expecting x-national variation could be sharpened by saying more about what factors would drive this.  Similarly, the individual-level hypotheses are all lumped together and don’t seem to capture much that is distinct to mothers.   The measure of underemployment has some limitations and generally needs a fuller justification. First, the measure taps “perceived” underemployment, which is distinct from measures that tap skill matches (e.g.,  more or less educated relative to job requirements).  Also, the decision not to use an “availability” criterion to determine underemployment needs further explanation.  Though it is likely true that lack of childcare (or costs of childcare) may limit some mothers’ availability for work, availability of other kinds may also affect non-mothers or men.  It would be useful to hear more about the lack of an availability criterion affected the degree of underemployment (or the results). 

        3a. A second issue with the underemployment measure is its treatment as a categorical variable.  This means that a person who wanted to work 1 more hour/day is treated the same as one who wanted to work 4 or more hrs/day.  The lack of attention to the perceived level of underemployment may hinder the ability to understand this issue.  And those who are satisfied with their work hours and those would prefer to work fewer hours are both categorized as “not underemployed.”  Some research shows that there are women who would prefer to work less than they are currently and by not distinguishing between these different types of work hour mismatches, an opportunity for understanding mothers’ labor market experiences is missed.  At a minimum readers should hear more about the distribution of work hours in the sample and the way this distribution maps onto the workers’ preferred work hours. 

         3b. A third issue concerns the sample selection, which includes all mothers who worked at least one hour the previous week.  Not knowing more about the distribution of work hours among the sample (and by country) it is hard to know how this broad definition of employment may have shaped levels of underemployment in the sample, but certainly it would be useful to have information about average work hours for a country (by gender).  Footnote 3 acknowledges x-national differences in women’s labor force patterns and says it needs to be taken into account, but it’s not clear where or how the paper does this.

There is a tendency to treat IVs as causes of underemployment when the direction of the effect could easily go the other way. For example, the paper seems to suggest (lines 359, 395) that being in a more economically insecure position leads to underemployment, but being underemployed may also be one cause of economic insecurity. 

Overall, the paper takes up an interesting topic, but requires more development in several key areas.

Reviewer 3 Report

The paper deals with an interesting and societally relevant topic and is overall well-written and executed. I have a couple of comments and suggestions that you may consider for further improving your manuscript.

Please take into account that EU and OECD have increasingly used employment rates rather than unemployment rates in recent decades to assess labor market  when motivating your study When defining underemployment, please justify why you do not impose a threshold on actual working hours (i.e., that only those not working full-time can be underemployed); if you prefer not to do that, please provide some sensitivity analyses or descriptive information on how many mothers feel underemployed despite a fulltime job L. 121-124: do individuals in deprived households always feel underemployed or perhaps rather underpaid? This relates to my point above and needs to be discussed. There is some more literature on underemployment that you may consider integrating in your paper, e.g., McKee-Ryan, F. M., & Harvey, J. (2011). “I Have a Job, But . . .”: A Review of Underemployment. Journal of Management, 37(4), 962–996. https://doi.org/10.1177/0149206311398134; in particular The literature in the section on involuntary part-time work should be updated (L. 77 -84 and section 3, findings section page 9ff.). The following overview article may direct you to some more recent studies: Hipp, Lena, Janine Bernhardt, and Jutta Allmendinger. "Institutions and the prevalence of nonstandard employment." Socio-Economic Review2 (2015): 351-377. Define “depth” of underemployment early on (already in abstract if possible as readers – as myself – may not be familiar with this term) You may reconsider the formulation of formal hypotheses. Hypothesis 1 is kind of pointless (“there is variation”); if anything, you should predict in which country cluster you expect to observe more underemployment than in others. Likewise, Hypothesis 2 incorporates many different elements and is therefore also not straightforwardly be sustained or rejected. One way to rephrase your empirical predictions would be “Based on the literature review, we expect to observe that underemployment  is correlated with the following individual-level characteristics.  Hypotheses 3 to 5 are plausible but require some more motivation. How did you deal with those who worked more hours than preferred in your analyses? Having them simply in the category “not underemployed” is a little dissatisfying and empirically problematic. This issue therefore requires action or at least discussion. Information on where the macro-level variables are derived from should be provided in the Table on page 7, too; in addition, the links to the database should be provided in the reference section Personally, I found the section on the bivariate correlations to be somewhat superfluous. As multilevel models are estimated, the author(s) could simply refer to the results of those. To avoid omitted variable bias, I would recommend to also include age and duration coverage as covariates in the multivariable analyses. In general, the recommendation for multilevel model is to estimate random slopes (as it is very unlikely that the relationships for all individual-level variables are the same in all countries). This is something else the author(s) may consider or at least discuss.

 Thank you for providing me with the opportunity to read the paper and all best for the revisions.

Round 2

Reviewer 2 Report

This revised paper attempts to address questions raised by reviewers and author has included more explanations and qualifications in several areas.  However, I'm not convinced the paper makes a substantial contribution and I'm not sure how to improve it.

On one hand, the paper argues that it extends previous research because it uses multilevel data to examine underemployment.  I agree that this could be useful.  However, because of significant limitations in the multilevel data, the paper can't rely much on the multilevel results but instead relies on the descriptive info about each country.  Many of the footnotes urge readers to interpret results "with caution" or note problems in the data that could affect the results.  This raises the question of whether the multilevel analyses add value.

Regarding the multilevel model itself, the paper does discuss whether the individual country-level vars can explain the variance at level 2, but they don't discuss how much variance in the model remains after the level 1 vars have been included.  In other words, they don't use their model to examine to what degree we can say that underemployment of mothers in this sample is driven mainly by individual-level factors or has a significant country-level component.  Ideally we would want to know if there are cross-level effects, but those can't be tested in these data.  

In short, I think the paper raises a lot more questions than it answers and am not sure that the multilevel approach can add much, given all the caveats about these data.